# A Novel Therapeutic Formulation for the Improved Treatment of Indian Red Scorpion (*Mesobuthus tamulus*) Venom-Induced Toxicity-Tested in *Caenorhabditis elegans* and Rodent Models

**DOI:** 10.3390/toxins15080504

**Published:** 2023-08-14

**Authors:** Bhabana Das, Dev Madhubala, Saurov Mahanta, Aparup Patra, Upasana Puzari, Mojibur R. Khan, Ashis K. Mukherjee

**Affiliations:** 1Microbial Biotechnology and Protein Research Laboratory, Department of Molecular Biology and Biotechnology, Tezpur University, Tezpur 784028, Assam, India; bhabana@tezu.ernet.in (B.D.); dmadhu@tezu.ernet.in (D.M.); upa2019@tezu.ernet.in (U.P.); 2Division of Life Sciences, Institute of Advanced Study in Science and Technology, Vigyan Path, Garchuk, Paschim Boragaon, Guwahati 781035, Assam, India; patra.aparup@gmail.com (A.P.); mojibur.khan@gmail.com (M.R.K.); 3National Institute of Electronics and Information Technology (NIELIT), Guwahati 781008, Assam, India; saurov.mahanta@gmail.com

**Keywords:** *M. tamulus* venom, neutralisation potency, formulated drug

## Abstract

Indian Red Scorpion (*Mesobuthus tamulus*) stings are a neglected public health problem in tropical and sub-tropical countries, including India. The drawbacks of conventional therapies using commercial anti-scorpion antivenom (ASA) and α1-adrenoreceptor antagonists (AAA) have prompted us to search for an adequate formulation to improve treatment against *M. tamulus* stings. Novel therapeutic drug formulations (TDF) of low doses of commercial ASA, AAA, and ascorbic acid have remarkably improved in neutralising the in vivo toxic effects of *M. tamulus* venom (MTV) tested in *Caenorhabditis elegans* and Wistar strain albino rats in vivo models. The neutralisation of MTV-induced production of free radicals, alteration of the mitochondrial transmembrane potential, and upregulated expression of genes involved in apoptosis, detoxification, and stress response in *C. elegans* by TDF surpassed the same effect shown by individual components of the TDF. Further, TDF efficiently neutralized the MTV-induced increase in blood glucose level within 30 to 60 min post-treatment, organ tissue damage, necrosis, and pulmonary oedema in Wistar rats, indicating its clinical application for effecting treating *M. tamulus* envenomation. This study demonstrates for the first time that *C. elegans* can be a model organism for screening the neutralization potency of the drug molecules against a neurotoxic scorpion venom.

## 1. Introduction

Scorpion sting, an under-researched prevalence, is responsible for many mortalities in most countries. The Indian red scorpion (*Mesobuthus tamulus*), with its life-threatening sting, is one of the world’s most dangerous scorpions [1]. The qualitative and quantitative occurrence of different venom toxins influences the pharmacological properties and toxicity of venom. Proteomic analysis has demonstrated the relative proportion of various toxins in the *M. tamulus* venom (MTV) [2]. Notably, the potent toxicity of MTV is attributed to the abundance of Na^+^ and K^+^ channel toxins (cumulative relative proportion is 76.7%), which target the central nervous system (CNS) and cardiovascular systems that result in clinical symptoms such as vomiting, sweating, salivation, bradycardia, tachycardia, pulmonary oedema, hypertension, acute pancreatitis, hyperglycaemia cardiac arrhythmias, etc. [2,3,4,5,6,7,8].

The neurologic symptoms associated with scorpion stings are varied and complicated, produced by various neurotoxic pathways, and are predominantly reported in children. The scorpion toxins stimulate the autonomic nervous system (ANS), both sympathetic and parasympathetic, triggering the coagulation cascade. The neurological symptoms are autonomic dysfunction, seizures, and ischemic or haemorrhagic stroke [9,10,11]. Toxins affect several voltage-gated ion channels, primarily Na^+^, K^+^, Cl^−^, and Ca^2+^. The toxin prolongs depolarization by acting on Na^+^ (excitation) and K^+^ (blocking) channels, primarily at the level of postsynaptic postganglionic nerve terminals of the ANS. This effect could result in a massive release of sympathetic (catecholamine) and parasympathetic (acetylcholine) mediators, resulting in a mixed neuroexcitatory syndrome [12,13]. *M. tamulus* (MT) stings cause symptoms such as ophthalmoplegia, ptosis, and respiratory failure in most patients due to neuromuscular dysfunction [13].

Intravenous administration of equine anti-scorpion antivenom (ASA), raised against MTV, is the preferred treatment for scorpion stings. However, the efficacy and quality of the ASA are significant concerns for successful antivenom therapy against scorpion stings [2]. Furthermore, the failure of commercial ASA to immunorecognise the most abundant low molecular toxins of MTV due to the presence of a low proportion of venom-specific antibodies in commercial ASAs is another hurdle for efficient hospital management of scorpion sting victims [2,14]. Therefore, a higher volume of ASA must be administered to scorpion sting patients, which can cause adverse serum reactions in treated patients [15].

Stimulation of the α1-adrenergic receptor by MTV plays a significant role in its pharmacology, resulting in clinical symptoms such as hypertension, tachycardia, myocardial dysfunction, pulmonary oedema, and cool extremities in patients [1]. Therefore, α1-adrenoreceptor agonists (AAAs), such as Prazosin, are also used alone or in combination with commercial ASA for treating scorpion stings [16]. However, treatment with AAAs (Prazosin) also has limitations; for example, it causes hyperglycaemia in patients at a higher dose [17,18]. MTV also causes hyperglycaemia in stung patients [19]. Consequently, AAA and MTV in unison would cause a sudden abrupt increase in blood glucose resulting in hyperglycaemic shock to the patients. For a diabetic patient, it would be a severe life-threatening consequence.

The effect of scorpion venom to induce tissue damage to organs (heart, liver, kidney, lung, etc.) owing to the generation of free radicals has been reported, and antioxidants are found to protect them from damage partially [15,16]. Notably, sepsis which is shown in patients in case of severe scorpion envenomation is a multi-organ dysfunction condition characterized by hyperinflammation, oxidative damage, hypercoagulation, tissue hypoperfusion and hypoxia, immunological suppression, and multi-organ malfunction [20].

A high dose of ascorbic acid or vitamin C (300 mg/kg) can decrease the toxic effect of the low-dose venom (1 to 3 µg/g; intra-peritoneal route) of *Hemiscorpius lepturus* scorpion (native to the deserts of the Middle East) on adult male Sarague–Dowley rats [21]. Nonetheless, the effect of ascorbic acid in decreasing the blood glucose level or reducing the damage to the vital organs, such as the liver and kidneys, at a higher scorpion venom dose has never been reported. Therefore, there is an immediate need to discover a potent combination of drugs that are more effective than conventional medications but devoid of adverse reactions in treating MT sting patients. 

This study demonstrated that a formulation drug comprising commercial ASA, AAA, and ascorbic acid is more efficient in vivo neutralising MTV-induced toxicity in two model organisms: *Caenorhabditis elegans* and Wistar strain albino rats, compared to individual components of the formulation. For drug screenings and therapeutic target discovery, *C. elegans* has been successfully employed to recapitulate a variety of human illnesses at the molecular level. Different octopamine receptor antagonist drugs can also be screened in *C. elegans* [22,23]. This nematode model also acts as an alternative model to mammalian systems, which can be used in pre-mammalian screening to save the lives of experimental animals [24,25,26]. Notably, this study is the first report showing that *C. elegans*, a nematode, can serve as a model organism for screening the neutralization potency of the therapeutic formulation/drug molecules against a neurotoxic scorpion venom.

The results of this study have highlighted that the therapeutic formulation demonstrated in this study may pave the way for the significant improvement of in-patient management of scorpion stings and eliminate the adverse effects of conventional treatments.

## 2. Results

### 2.1. The In-Silico Analysis Showed the Binding of AAAs to Homologous SER6 Receptors in C. elegans

A comparison of the in-silico binding efficiency of AAAs to α-adrenergic receptors of humans, mice, and homologous receptor (SER6 receptor) in *C. elegans* is shown in Appendix A. Prazosin-HCL demonstrated the highest binding efficiency to the α1A adrenergic receptor of humans. In contrast, Silodosin and Terazosin-HCL showed the highest binding efficiency to α1D-adrenergic receptors in mice and SER6 receptor in *C. elegans*, respectively (Appendix A, Appendix A).

A molecular docking study showed that Prazosin-HCL binds with AAA via one hydrogen bonding and binds to Gln 345 of the SER6 receptor in *C. elegans*. In contrast, Terazosin-HCL showed binding via one hydrogen bonding to Pro 106 and Silodosin via three hydrogen bodings to Gln 345, Gln 154, and Thr 363 of the same receptor (binding affinity). The binding relationship of Prazosin-HCL, Terazosin-HCL, and Silodosin with respective receptors was predicted at −6.6, −7.5, and −6.8 kcal mol^−1^, respectively (Appendix A).

### 2.2. Optimum Dose of Inhibitors in Neutralizing the MTV-Induced Lethality, ROS Generation, and Depolarization of Mitochondrial Transmembrane Potential in C. elegans

The commercial ASAs (PSVPL and HBC) (2 mg), AAAs (100 µM), and ascorbic acid (2 µg) did not show toxicity in *C. elegans* (Appendix A). The regression analysis determined the LC_50_ value of MTV towards *C. elegans* after 24 h of incubation as 125 µg/mL (Appendix A).

The present study provides new information on the in vivo MTV neutralisation efficacy of ASAs, AAAs, and ascorbic acid in the *C. elegans* model. All the tested inhibitors, except varespladib, showed dose-dependent neutralisation MTV-induced toxicity (in terms of survivability) in *C. elegans*; however, to a significantly (* *p* ≤ 0.05) different extent. The optimum MTV neutralisation dose of ASA, AAA (Prazosin in this study), and ascorbic acid determined are shown in Appendix A. The AAAs showed better neutralisation potency than commercial ASAs and ascorbic acid (Figure 1a–e) against MTV-induced lethality in *C. elegans*. However, Silodosin showed better MTV-neutralisation potency than Prazosin and Terazosin (Figure 1b–d). Further, PSVPL ASA was slightly more effective (* *p* ≤ 0.05) than HBC ASA in neutralising the in vivo toxicity of MTV in C. elegans (Figure 1a). Interestingly, ascorbic acid (1 µg) and ASAs (1500 µg) showed equipotency in neutralising the MTV-induced toxicity in *C. elegans* (Figure 1e). 

The ROS production in *C. elegans* after 6 h incubation with LC_50_ concentration of MTV was significantly neutralised in the following order: AAAs > ascorbic acid ≥ commercial ASAs (Figure 2a; Appendix A). The confocal laser microscopic study demonstrated that MTV time-dependently increased the mitochondrial membrane depolarisation in *C. elegans*, significantly neutralised by AAAs followed by ascorbic acid and commercial ASAs (Figure 2b, Appendix A).

### 2.3. Early Treatment with AAAs, Commercial ASAs, and Ascorbic Acid Showed Better Neutralisation of MTV-Induced Toxicity in C. elegans

When *C. elegans* were treated with an optimum dose of AAAs (25 µM for Silodosin and 50 µM for both Prazosin and Terazosin) and commercial ASAs (1500 µg) at 0, 30, 60, and 120 min post-MTV treatment, percent viability of worms gradually decreased (Figure 3a–e). These results suggested early administration (within 30 min post-MTV addition) of AAAs or ASA to scorpion sting patients for better management of the post-envenomation effect. Notably, primary treatment with ascorbic acid (60 and 120 min before the addition of ASAs) compared to treatment with ASAs at 60 and 120 min post-addition of MTV was found to be beneficial for efficient therapy of MTV-induced toxicity in *C. elegans* (Table 1).

### 2.4. Formulated Drug Showed Significantly Higher Efficiency Compared to Individual Components of Formulation in Neutralising the MTV-Induced Lethality in C. elegans

The neutralisation potency of different formulation concentrations, viz formulation 1, formulation 2, and formulation 3 have been evaluated against the lethality of MTV induced *C. elegans*. Formulation 1 shows lower potency than formulation 2; however, formulation 3 did not give any additional advantage (*p* > 0.05) as compared to formulation 2 in neutralising the MTV-induced toxicity in *C. elegans* (Table 2). Further, formulation 2 demonstrated significantly higher efficiency than individual formulation components and their combinations in neutralising the MTV-induced lethality in *C. elegans* and was considered for further studies (Table 2).

### 2.5. The Formulated Drug (Formulation 2) Demonstrated Optimum Efficiency in Neutralising the In Vitro DPPH-Free Radical Scavenging Activity and In Vivo Neutralisation of MTV-Induced Reactive Oxygen Species (ROS) Generation and Alteration of Mitochondrial Transmembrane Potential (MMP) in C. elegans

Formulations 2 and 3 compared to formulation 1 and individual components demonstrated higher in vitro DPPH-free radical scavenging activity (Appendix A). There was no significant difference in free-radical scavenging activity between formulations 2 and 3, meaning that formulation 2 has effective free-radical scavenging activity.

At 6 h of incubation, MTV-induced ROS production was significantly reduced by formulations 2 and 3 (Figure 4a–c). However, there was no significant difference (*p* > 0.05) in the inhibition of ROS generation between formulation 2 and formulation 3, but their inhibitory potency was significantly higher compared to formulation 1 (Figure 4a).

Formulations 2 and 3 significantly decreased MTV-induced increase in the MMP of *C. elegans* compared to formulation 1 and the individual components of the formulations and their combinations (Figure 5a–c). However, there was no significant difference in potency between formulations 2 and 3 (Figure 5a).

### 2.6. Formulated Drug Restored the MTV-Induced Upregulation of Genes Involved in Apoptosis, Detoxification, and Stress Response to Delay MTV-Induced Programmed Cell Death in C. elegans

Because the composition of formulation 2 was found to be optimum for neutralising the MTV-induced toxic effects in *C. elegans*, further studies were done with these formulations. The expressions of ctl-1 and gst-6 (anti-oxidative genes) were significantly upregulated in MTV-induced *C. elegans* compared to the control (untreated). The expression of hsp60 (heat shock gene) was also upregulated considerably; however, the expression of sod-3 and trx-1 was unchanged in MTV-treated *C. elegans* (Figure 6). The qRT-PCR results showed that the MTV-induced expression of upregulated genes was downregulated to normal by formulation 2 (Figure 6).

### 2.7. Neutralisation of MTV-Induced Hyperglycemia and Pathophysiological Symptoms, Prolonged Tail Bleeding Time, Serum Biochemical Changes, and Morphological Alterations in Wistar Strain Albino Rats Model by Drug Formulation 2

As shown in Appendix A, MTV-treated Wistar strain albino rats became physically inactive and almost paralyzed after 24 h of treatment. They showed high urination, fast breathing, defecation, becoming thirstier, and weak grip strength. However, the formulated drug (formulation 2) restored the regular physiological activity of MTV-treated Wistar strain albino rats.

A significant increase (* *p* ≤ 0.05) in the blood glucose level in MTV-treated rats compared to the control group of rats was observed post 30 to 120 min i.v injection of venom (Figure 7a). The AAA at its optimum dose (determined in *C. elegans*) markedly increased the glucose content at 30 to 120 min post-injection (Figure 7a). Moreover, the quantity of the combination (two components at a time) of formulation 2 did not effectively lower MTV-induced hyperglycaemia as compared to formulation 2 (^¥^
*p* ≤ 0.05) in Wistar strain albino rat suggesting the optimum composition of formulation 2 as the most effective formulation to diminish the hyperglycaemic effect (Figure 7a).

The tail bleeding time was also prolonged in Wistar strain albino rats post-treatment with MTV, which was neutralised by formulation 2 (Figure 7b). A similar effect was also observed when treated with an optimum dose of individual components and a combination dose of formulation 2 (^¥^
*p* ≤ 0.05), the later showed lower potency in neutralising the MTV-induced prolonged tail bleeding time suggesting that formulation 2 but not its individual components are the most efficient in the treatment of MT stings (Figure 7b).

The MTV-treated rats showed increased serum SGPT, ALKP, and creatinine levels compared to the control group of rats. The MTV-induced increased liver enzymes (ALKP and SGPT) and metabolic product (creatinine) were better reduced by formulation 2 (Table 3). Further, the study also suggested that formulation 2 is effective compared to individual components of the formulation for restoration of MTV-induced serum biological changes (Table 3).

Histological analysis of the MTV-treated rats’ hearts, kidneys, livers, and lungs revealed some gross morphological alterations (Appendix A). MTV-induced heart muscle showed massive deleterious degeneration, and almost no intact muscle tissue remained (Appendix A). The kidney showed some black spots on the tissues along with necrosis (Appendix A), and the liver also suffers from tissue necrosis (Appendix A). Inflammation with pulmonary oedema was also observed in MTV-induced lung tissues filled with proteinaceous fluid (Appendix A). Ovary and testis of the MTV-induced rat did not show any morphological change compared to the control (Appendix A). The toxic effect of MTV in a rat model was neutralised when treated with formulation drug 2 (Appendix A).

### 2.8. Decrease of Pro-Inflammatory Cytokines in MTV-Treated Swiss Albino Mice

The level of pro-inflammatory cytokines in the MTV-treated Swiss albino mice was significantly decreased compared to the control mice (Appendix A).

## 3. Discussion

In vivo, toxicity studies with mammalian models are expensive [27,28] and have ethical concerns. Nevertheless, the prediction of toxicity study can be augmented by using more than one mammalian species, but it will also increase the cost and moral significance and decrease throughput [29]. To overcome these limitations, for studying the toxic effects of scorpion venom and screening the venom neutralisation potency of drugs, *C. elegans*, a tiny non-parasitic nematode, is one of the best-established in vivo models that have contributed significantly to understanding numerous human diseases such as neurodegenerative disease, diabetes, etc. [26,30]. It also serves as a model organism for genetic studies on the aging process, age-related illnesses, mechanisms of longevity, and screening for compounds that increase lifespan [30,31]. It is a very striking experimental model due to many advantages, such as its small size and short life cycle (approx. three days at 20 °C), ability to self-fertilize, and high reproductive rate (>300 offspring per hermaphrodite). Different studies throughout the globe have already shown that *C. elegans* is a good choice for studying the neurotoxic effects of toxic chemicals (such as pesticides) [32,33].

Considering the above, in our study, *C. elegans* has been used as a model organism to study the neurotoxic effects of MTV and the screening of the neutralization potency of a novel therapeutic formulation against MTV-induced toxicity. The formulated drug has successfully inhibited the mammalian pathophysiology associated with neurotoxic symptoms, such as free radicals scavenging, ROS inhibition, and inhibition of depolarization of MMP, resulting from MTV envenomation. However, we agree that *C. elegans* is not the final model for establishing the therapeutic efficacy of drug molecules, instead of that our aim of the study is to find an alternative in vivo screening model to solve the ethical issue associated with the animal experiment and also to follow 3R model (Replacement, Reduction, and Refinement of tested animals) suggested by WHO (World Health Organization) to reduce the animal experiments. Further, the best-screened formulation was validated in rodent models where the formulated drug has shown superior efficacy compared to the individual components of the same drug to neutralize the toxicity of Indian red scorpion venom, thus establishing the suitability of C. *elegans* model to screen drugs against MTV. This in vivo model may also be used to screen drugs against other scorpions and neurotoxic venoms. 

The hyperactivity of α-adrenergic receptors by scorpion venom toxins leads to cardiovascular effects with a first phase of transient cholinergic hyperfunction, followed by dose-dependent adrenergic hyperactivity characterized by hypertension, tachycardia, and alteration of myocardial function. Along with the cardiovascular effect, hyperactivity of the α-adrenergic receptor activates the neurotransmitter norepinephrine and the neurohormone epinephrine, also known as catecholamines. The massive release of sympathetic (catecholamine) post-scorpion sting results in a mixed neuroexcitatory syndrome [13,34]. Therefore, assessing the neurotoxicity-related pathophysiology such as ROS generation and disruption of the mitochondrial membrane potential of scorpion venom in *C. elgans* has a rationale. However, there are some limitations to using *C. elegans* as a model organism, such as studying scorpion sting-induced cardiovascular alterations. Nevertheless, *C. elegans* also exhibit limitations as they lack many mammalian organs such as eyes, lungs, heart, kidney, and liver and are devoid of an adaptive immune system. Despite these, they are considered good toxicity testing models that help bridge between in vitro assays and mammalian toxicity studies [35]. Therefore, the scorpion venom-neutralizing potency of screened drug molecules must be validated in pre-clinical studies. 

Notably, *C. elegans* also poses a homologous receptor of the mammalian α1-adrenergic receptor, SER6 [36] which is demonstrated by in-silico analysis in this study. SER6 is an octopamine receptor, and its stimulation causes an octopaminergic signal that involves an array of neuropeptides that activate receptors and induce a cAMP response in the CNS, thus showing pathophysiology associated with neurotoxicity such as ROS generation and disruption of mitochondrial membrane potential [37,38].

Therefore, at the outset, we tested the MTV neutralisation potency of formulated drugs (formulation 1, 2, and 3) and individual components of the formulation in *C. elegans* to save the experimental animals. This study was followed by determining the MTV toxicity neutralising potency of the formulated drug (best formulated drug after screening in *C. elegans*) in a rodent model (Wistar strain albino rats).

Stimulation of α1-adrenergic receptors (a classic post-synaptic α-receptor found on vascular smooth muscle) by scorpion venom α-neurotoxins induce hyperkalaemia and hyperglycaemia, accumulation of free radicals in the myocardium, and coronary spasm and thus initiating lethal ventricular arrhythmias and sudden death [39]. Most clinical manifestations post scorpion sting envenomation result from the massive release of neurotransmitters such as catecholamines leading to autonomic storm [40,41]. Sepsis, a multi-organ dysfunction condition in extreme cases, is caused by a combination of immune cell malfunction (macrophages, neutrophils, and lymphocytes), endothelial cell dysfunction, and epithelial cell failure. ROS and reactive nitrogen species (RNS) both contribute considerably to these cells’ dysfunction during sepsis [20]. The balance of pro- and anti-inflammatory activity determines the intensity and degree of inflammation, which results in various clinical outcomes. Cytokine imbalances induced by scorpion venom toxins play a role in developing organ damage and fatality during severe sepsis. Although the sepsis induced by scorpion sting cannot be assessed in *C. elegans*; however, ROS generation and alteration of mitochondrial transmembrane potential, the primary causes of sepsis, can be determined in *C. elegans*. We have demonstrated that the formulated drug effectively inhibits ROS production and restores the disruption of MMP. For the management of sepsis, along with antivenom treatment, antibiotic treatment is suggested [42,43,44]. Further, few studies have also reported that the use of prazosin to mitigate sepsis in scorpion-stung patients. Furthermore, a histopathological study in rodent models has shown a deleterious effect of MTV in different organs of rats and its effective neutralization by formulated drug, much better than the individual components of the drug. The studies of marker enzymes in the serum of MTV-treated rats showed an increase in SGPT, ALKP, and creatinine indicating damage in the liver and kidney. Our formulated drug restores the level of marker enzyme significantly higher than the individual components. Nevertheless, an in-depth study into the mechanism of MTV-induced sepsis and its neutralization is utmost needed.

Hyperglycaemia, one of the scorpion venom-induced symptoms, is caused by an increase in counter-regulatory hormones that inhibit gluconeogenesis and glycogenolysis, leading to respiratory failure, multisystem organ failure, pulmonary oedema, and increased mortality rate [19,45,46,47]. Prazosin, a post-adrenergic receptor blocker with 1000-fold more affinity to the α1-adrenergic receptor, has been clinically used with commercial ASAs to reduce a post-scorpion sting’s pharmacological and toxic effects [39].

Before initiating the wet lab experiment, the in silico analysis was done to predict whether or not the AAA could bind to the SER6 receptor of *C. elegans*. The computational analysis predicted the efficient binding of AAA with the homologous SER6 receptor indicating that *C. elegans* is a good model for assaying the effect of AAAs against scorpion venom. Our present study revealed that all tested AAAs showed more efficient neutralisation than commercial ASA against MTV, which is in close agreement with the previous clinical data showing Prazosin as an effective drug compared to scorpion antivenom [48]. Scorpion venom induces apoptosis and cell death by elevating reactive oxygen species (ROS) [49]. Thus, this study has presented evidence that MTV also generates free radicals, which induce the production of ROS, the oxidative stress response, and alter the mitochondrial trans-membrane potential in *C. elegans*. Our present study revealed that MTV-induced toxicity (increased ROS production and depolarisation of mitochondrial transmembrane potential) was better neutralised by AAA than commercial ASACommercial ASA generally neutralizes scorpion venom toxicity [50,51]. However, our previous study has revealed that a high concentration of ASA (1:60; venom: antivenom) was required to neutralise (in vitro) the toxicity of MTV; the reason may be due to the presence of a low abundance of venom-specific antibodies (5.36–6.29%) in commercial ASAs [14]. Due to their insufficient venom-specific antibodies and some adverse side effects (e.g., hypersensitivity, anaphylactic reactions, serum sickness, etc.) [15], the commercial ASA may prevent scorpion sting patients from recovering completely, suggesting improving scorpion sting management protocol. Furthermore, Prazosin at higher concentrations used to treat Indian red scorpion stings also induces hyperglycaemia [17]. Our present study also evidenced that MTV-induced hyperglycaemia was not reduced when treated with prazosin (50 µM); instead, it raised glucose levels in rat serum. The sudden increase in blood glucose can produce a hyperglycaemic shock in the patients. It is a severe life-threatening problem for diabetic patients. 

The therapeutic advantages of early administration of AAAs, or ASA, for effective treatment of scorpion stings and consequently saving a patient’s life are reinforced in this study. ASA is only sometimes available in most rural health centres in developing countries, and patients must travel a long distance to get the treatment [52]. Therefore, based on the results of these studies, we proposed that immediate administration of ascorbic acid can delay the onset of pharmacological effects post-scorpion envenomation, and the patient can get sufficient time to reach a hospital where an envenomation treatment facility is available. In this study, both the models (*C. elegans* and Wistar strain rats) demonstrated improved neutralisation efficiency of the formulated drug 2 (composed of ASA, AAA and ascorbic acid) against MTV-induced toxicity by increasing the percent worm survivability rate, lowering free radical production in *C. elegans.* We showed that MTV causes upregulation of genes involved in apoptosis, detoxification, and stress response in *C. elegans* after 24 h of MTV treatment. The formulated drug stored this upregulation of *C. elegans* genes mentioned above.

Scorpion venom causes hyperglycaemia via different mechanisms such as massive catecholamine release, glucagon release, and glucocorticoid release, affecting renin–angiotensin–aldosterone system, causing hyperinsulinemia and cytokines release, etc. [53,54]. However, the precise mechanism of MTV-induced hyperglycaemia is unknown. In our present study, drug formulation 2 also envisaged reducing MTV-induced hyperglycaemia in Wistar strain albino rats, much better than commercial ASA or ascorbic acid. It also improved the pathophysiological changes and altered tissue morphological deformation caused by MTV toxicity in Wistar strain albino rats. With formulation 2, insulin treatment may not be required for hospital management of scorpion sting patients.

Clinical reports of MTV-induced rat serum indicated a significant increase of liver enzymes such as ALKP and SGPT; creatinine (an indicator of kidney function) after 24 h post-treatment. However, the SGOT (ALT) level did not change compared to the control group. Studies have shown the rise in serum levels of SGOT, SGPT (AST), and/or glucose, cholesterol, uric acid, bilirubin, and urea in experimental animals’ post-treatment with scorpions (*Hemiscorpius lepturus*, *Odonthobuthus doriae*, *Androctonus crassicauda*, and *Palamneus gravimanus)* venom samples [55,56,57,58]. In this study, formulated drug 2, compared to its components, has also shown an impressive result in lowering MTV-induced increased biochemical parameters such as ALKP, SGPT, and creatinine in Wistar strain albino rats suggesting its therapeutic application in treating MT sting. 

MTV-treated rats showed histological symptoms, including tissue necrosis, oedema, and muscular injury observed in tissue such as the heart, kidney, liver, and lung. In our study, myocardial damage was observed in rat heart muscle. Scorpion venom-induced myocardial damage has also been reported as one of the major clinical manifestations that may lead to death [59]. The present study also revealed the disorganization of liver tissue with a necrotic lesion; many experimental studies have also reported histopathological changes in the liver caused by scorpion sting envenomation which may lead to toxic hepatitis and coagulopathy [60]. Pulmonary oedema is a common and significant systemic clinical manifestation, which was also observed in MTV-treated rat lung tissue [61]. These morphological alterations were diminished after treatment with the most effective formulated drug 2, which helped to reverse the imbalance caused by MTV without causing adverse reactions.

Many cardiogenic and non-cardiogenic factors are involved in the pathogenesis of acute pulmonary oedema after scorpion stings [62,63]. But no such effect was observed on MTV-induced rat testis and ovary tissue. The study of in vivo neutralization of MTV-induced toxicity by the formulated drug in *C. elegans* was well correlated with results with the Wistar strain albino rats’ model, supporting the proposed hypothesis that *C. elegans* can also serve as a model to screen the antidotes against scorpion venom. 

## 4. Conclusions

Sustainable availability of tremendous and safe antivenom immunoglobulins must be ensured, and manufacturing systems for these effective remedies should be strengthened globally. The pre-clinical assessment for assessing the efficacy of antivenom and other drugs serves as a gold standard by determining their neutralisation potency against scorpion venom-induced toxicity. Further, from a public health forthcoming, the modern approach to antivenom development needs to be improved to reduce complications and better manage scorpion sting envenomation. Thus, our study presented the efficacy of a formulated drug comprising low doses of commercial ASA, AAA, and ascorbic acid, capable of neutralising MTV’s in vivo toxic effect in *C. elegans* and Wistar strain albino rats. Both models exhibited significantly improved efficiency of the developed drug compared to its component against MTV-induced toxicity.

## 5. Materials and Methods

### 5.1. Chemicals and Reagents

Lyophilised MTV was a gift from PSVPL, Pune, India. Lyophilised commercial ASAs were procured from Haffkine Bio-pharmaceutical Corporation Ltd. (HBC), Mumbai, India (batch No.: PRMSC-002, expiry date: February 2022), and Premium Serum and Vaccines Pvt. Ltd. (PSVPL), Pune, India (batch No.: SS170401, expiry date: September 2021). α1-adrenoreceptor antagonists (AAA) such as Prazosin, Silodosin, and Terazosin were obtained from Aristo Pharmaceuticals Pvt. Ltd., Mumbai, India; Sun Pharma Laboratories Ltd., Mumbai, India; and Abbott India Ltd., Himachal Pradesh, India, respectively. The *Caenorhabditis* Genetics Center (CGC), University of Minnesota, USA, provided *C. elegans* wild-type strain N_2_. The laboratory inbreeds, pathogen-free Wistar strain albino rats (180–200 g), and Swiss albino mice (18–20 g) were purchased from M/S Chakrabarty Enterprise, Kolkata, and were used for the experiments. The animal experiment was approved by the institutional animal ethics committee (IASST/IAEC/2022/09), and experiments were performed following OECD guidelines. The present work has been applied for a patent (Application number: 202331027004). Mito Probe TM JC-1 Assay kit (Cat-M34152) and pure link RNA mini kit (Cat-12183018A) were obtained from Invitrogen, Carlsbad, CA, USA. Verso cDNA Synthesis Kit (Cat-AB-1453/A) was purchased from Thermo Scientific, Lithuania, USA. Mouse IL-1β/IL-1F2, IL-6, and TNFα Quantikine^®^ Mouse Immunoassay kits (Cat-MLB00C, M6000B, and DY410, respectively) were purchased from Biotechne R and D systems, Inc. (Canada, USA). All other chemicals were from Sigma-Aldrich, Louis, MO, USA and HIMEDIA, Maharashtra, India.

### 5.2. Computational (In Silico) Analysis to Compare the Binding Efficiency of AAAs between α1-Adrenergic Receptor (α1A, α1B, and α1D) in Humans and Mice and Homologous Receptor (SER6) in C. elegans

As the 3D protein structures for the selected proteins were not available in RCSB (Research Collaborator for Structural Bioinformatics) Protein Data Bank, the protein structures were taken from the predicted structure database of Alpha Fold [64,65], where Deep Mind has applied artificial intelligence (AI) driven methods to solve the protein structures accurately. The Alpha Fold identifiers and the Swiss-Prot Accession numbers have been mentioned in Appendix A. The Alpha Fold predicted structures were docked against the ligands (drug molecules), viz., Prazosin-HCL, Terazosin-HCL, and Silodosin using AutoDock Vina [66]. 

#### 5.2.1. Preparation of the Ligand 3D Structures for Docking

The 2D structures of the ligands to be docked, viz. Prazosin-HCL [PMCID: 68546], Silodosin [PMCID: 5312125], and Terazosin-HCL [PMCID: 44383] were downloaded from the PubChem Compound Database of NCBI. 3D models of the ligands were generated using CORINA classic 3D software [67,68] and subjected to energy minimization using UCSF Chimera [69].

#### 5.2.2. Protein-Ligand Docking

The AutoDock Vina, a widely used and fast Open source program for molecular docking, was used for this study through the Python-based PyRx virtual screening platform [70]. 

### 5.3. Determination of In Vivo Neutralisation Potency of Commercial ASAs, AAAs, and Ascorbic Acid in C. elegans Model

#### 5.3.1. Cultivation and Synchronization N_2_
*C. elegans* Worms

Wild-type N_2_ nematodes were kept in Petri dishes containing nematode growth medium (NGM, composed of 3.0 g NaCl, 2.5 g peptone, 17 g agar, autoclaved, and supplemented with 24 mL phosphate buffer, 1 mL 1 M CaCl_2_, 1 mL 1 M MgSO_4_, and 1 mL of 5 mg/mL cholesterol dissolved in ethanol) [71] and fed with OP_50_
*Escherichia coli*. The synchronised L1 larvae of N_2_ nematodes were transferred to the NGM containing *E. coli* OP_50_ at 20 °C for 48 h. Then late L4-young adult stage worms were transferred to a 50 mL sterile conical centrifuge tube and spun for 2 min at 1150× *g* to pellet the worms. After aspirating the remaining liquid, worms were transferred to 250 mL of S Basal inoculated with *E. coli* OP_50_. The worms were monitored by checking a drop of the culture under the stereomicroscope (Labomed CZM-4). Mid-L1, mid-L2, mid-L3, and mid-L4 larvae were harvested after approximately 8, 18, 25, and 37 h, respectively, at 20 °C [72].

#### 5.3.2. Determination of Lethal Concentration 50 (LC_50_) of MTV in *C. elegans*

For determining the LC_50_ value (the concentration of MTV at which 50% of the larvae died), L4 stage N_2_ worms (50 worms/well) were grown in NGM containing graded concentrations of MTV (33.3 to 500 µg/mL) for 24 h at 20°C. The control larvae were raised in NGM only. The worms were fed with *E. coli* OP_50_ ad libitum and observed for survival and paralysis/death under a stereo-zoom microscope. Those worms that showed no movement after exposure to light and gentle tapping were considered dead or paralysed. Notably, the paralysis induced by MTV was irreversible. The LC_50_ value was calculated from regression analysis of the dose–response curve by comparing dead or paralysed worms with controls. 

#### 5.3.3. Determination of Dose- and Time-Dependent Neutralisation of MTV-Induced Toxicity in *C. elegans* by Commercial ASAs, AAAs, and Ascorbic Acid 

The toxicity of a fixed concentration of commercial ASA (2 mg), AAAs (100 µM), and ascorbic acid (2 µg) in *C. elegans* (50 worms per well) was determined as described in Section 5.3.2.

Different amounts of commercial ASA (375 to 1500 µg) were pre-incubated with 125 µg/mL (LC_50_ value) of MTV (venom: antivenom; protein: protein) for 30 min at 37 °C, and the mixtures were added to synchronised late L4-young adult stage nematodes into the 48-well plates (each well contains 50 worms/200 µL reaction mixture); the wells containing *C. elegans* with NGM were considered as controls. The worms were kept at 20 °C, and their survival and paralysis/death were determined at 24 h post-treatment by eye quantification using a stereo-zoom microscope. 

In another set of experiments, different concentrations (6.25 to 50 µM) of AAAs viz. Prazosin, Silodosin, and Terazosin, and ascorbic acid (vitamin C) (0.2 to 2.0 µg) were mixed with LC_50_ concentration of MTV, and the mixtures were transferred to 48-well plates containing 50 worms/well. Veraspladib (400 µM), a known inhibitor of the PLA_2_ enzyme of venom [73], was used as a negative control because MTV lacks the PLA_2_ enzyme [2]. The survival and paralysis/death were determined at 24 h post-treatment as described above. The result was expressed as the percent viability of worms in control and treated groups at each time interval. The neutralisation potency of commercial ASA, AAAs, and ascorbic acid was determined.

For determining the time-dependent neutralisation of MTV-induced toxicity in *C. elegans*, the optimum concentration of each inhibitor (as determined from the results of the above experiments), viz. commercial ASAs, AAAs, and ascorbic acid was added to the culture of *C. elegans* (50 worms/200 µL reaction mixture) at 0 min (immediately after venom treatment) to 120 min post-treatment with MTV (LC_50_ value). The survival and paralysis/death of the worms were determined 24 h post-treatment, as described above. 

#### 5.3.4. In Vivo Neutralisation of MTV-Induced Generation of ROS and Alteration of MMP in *C. elegans* by ASA, AAAs, and Ascorbic Acid

For determining the neutralisation of ROS generation, the late L4-young adult stage nematodes were transferred into the 48-well plates (containing 50 worms in each group) and then treated with (i) LC_50_ value of MTV, the (ii) reaction mixture of MTV (LC_50_ value) and optimum dose of AAAs/commercial ASAs/ascorbic acid, and (iii) NGM (control)/ carbonyl cyanide m-chlorophenylhydrazone (CCCP; positive control). ROS level in the positive control (CCCP1) *C. elegans* was considered baseline (100%), and other values were compared. The worms were fed with *E. coli* OP_50_ and incubated for 6 h at 20 °C. Afterward, worms were washed twice with 1XM9 buffer, incubated with a fluorogenic probe 2′,7′-dichlorofluorescein-diacetate H_2_DCFDA stain at 37 °C for 5 h in the dark, followed by washing twice with 1XM9 buffer. Then, worms were placed on agarose bed slides and covered with a cover clip. Then fluorescence intensities of 2′,7′-dichlorodihydrofluorescein (DCF) produced by intracellular ROS were analysed by a confocal laser microscope (TCS SPE, Leica, Wetzlar, Germany) with excitation and emission wavelength at 480 nm and 530 nm, respectively [74].

The MTV-induced MMP was determined using 5,5′,6,6′-tetrachloro-1,1′,3,3′-tetraethylbenzimidazolylcarbocyanine iodide (JC-1) dye. Late L4-young adult stage *C. elegans* (50 worms) were treated with MTV (LC_50_)/NGM (control)/carbonyl cyanide m-chlorophenylhydrazone (CCCP; positive control)/MTV (LC_50_) mixed with an optimum dose of AAAs/commercial ASAs/ascorbic acid. After 12 h at 20 °C, the worms were washed twice with 1XM9 buffer, incubated with JC-1 with stain at 37 °C for 5 h in the dark, and then washed twice 1XM9 buffer. Then, worms were placed on agarose bed slides and covered with a cover clip. The fluorescence intensities were determined using a confocal laser microscope (Leica DMi8) with an excitation wavelength of 488 nm and emission wavelength of 533 ± 30 nm (FL-1 green channel) and 585 ± 40 nm (FL-2 red channel). The alteration of transmembrane potential by positive control (CCCP1) and crude venom was considered baseline, and other values were compared to that.

### 5.4. The In Vivo Neutralisation of MTV-Induced Lethality in C. elegans with Individual Components of the Formulation and Their Combinations

Synchronised nematodes were treated with individual components of formulation such as commercial ASA (PSVPL), AAA (Prazosin), ascorbic acid (vitamin C), their combinations, and different concentrations of the formulation against LC_50_ (125 µg/mL) value of MTV. The compositions of the components of the formulations are described in Appendix A. The percent survivability of *C. elegans* was determined as described above (Section 5.3.2). 

### 5.5. In Vitro DPPH Free Radical-Scavenging Activity of Different Concentrations of the Formulated Drug, Individual Components of the Formulation, and Their Combinations

DPPH (2,2-diphenyl-1-picrylhydrazyl) radical-scavenging activity was measured according to the method of Chen et al. (2020) [75]. Briefly, a 100 μL sample of the formulated drug, individual components of the formulation, and their combinations were mixed with a DPPH radical solution of 100 μL (2 × 10^−4^ mol/L, dissolved in 95% ethanol) in a 96-well plate. Then the solution was incubated for 30 min at room temperature. The absorbance of the solution at 517 nm was immediately measured by the Enspire microplate reader (Perkin Elmer, Inc., Baesweiler, Germany). The percentage of DPPH free radical-scavenging activity of formulated drug and components of the formulation was calculated as:DPPH radical scavenging activity (%): [1 − (A_1_ − A_2_)/A_0_] × 100,(1)
where A_0_ represents the absorbance of 100 μL of 95% ethanol (*v*/*v*) with a DPPH radical solution of 100 μL at 517 nm, A_1_ the absorbance of the sample (100 μL) with DPPH solution (100 μL), and A_2_ represents the absorbance of the sample (100 μL) solution with 95% ethanol (100 μL).

### 5.6. In Vivo Neutralisation of MTV-Induced Generation of ROS and Alteration MMP in C. elegans by Different Concentrations of the Formulated Drug, Individual Components of the Formulation, and Combination Thereof

For determining the neutralisation of ROS generation, the late L4-young adult stage nematodes were transferred into the 48-well plates (containing 50 worms in each group with 200 µL reaction mixture) and then treated with (i) LC_50_ value MTV, (ii) reaction mixture of MTV (LC_50_ value) and the individual component of the formulation and different concentrations of formulation, and (iii) NGM (control). MTV-induced ROS production was determined as described above in Section 5.3.4.

The MTV-induced MMP was determined using 5,5′,6,6′-tetrachloro-1,1′,3,3′-tetraethylbenzimidazolylcarbocyanine iodide (JC-1) dye. For the assessment of neutralisation MTV-induced alteration of MMP, late L4-young adult stage *C. elegans* (50 worms) were treated with MTV (LC_50_)/NGM (control)/CCCP (positive control)/MTV (LC_50_) mixed with components of the formulation and different concentrations of the formulation. After 12 h, fluorescence intensities were determined, as mentioned in Section 5.3.4.

### 5.7. Restoration of MTV-Induced Expression Level of Genes Involved in Apoptosis, Detoxification, and Stress Response by the Formulated Drug Was Determined by Quantitative Reverse Transcription-Polymerase Chain Reaction (qRT-PCR) 

Synchronised L4-young adult stage nematodes (approximately 500 worms) were treated with LC_50_ dose (125 µg/mL) of MTV/ NGM (control group) and incubated at 20 °C for 24 h [1]. After 24 h of treatment, worms were washed, followed by total RNA extraction using a pure link RNA mini kit. The RNA’s purity and concentration (A260/A280) were measured using a nanodrop spectrophotometer. Then c-DNA was synthesised with 1 µg of extracted RNA using a Verso c-DNA Synthesis Kit. To determine gene expression, q-RT-PCR was performed with the help of SYBR Green (Applied Biosystems, Foster City, CA, USA) in a real-time PCR machine (Applied Biosystem, Foster City, CA, USA). The amplified genes were pro-apoptotic (ced-3 and ced-4), anti-apoptotic (ced-9), detoxification GSTs (gst-6, gst-7), catalases (ctl-1), Thioredoxin-1 (trx-1) and SOD (sod-3), and heat-shock proteins HSP (hsp-60). Each gene’s relative expression was examined using the 2^−∆∆Ct^ method [74,76]. The housekeeping gene act-1 was used to normalise each gene expression.

### 5.8. Validation of In Vivo Neutralisation of MTV-Induced Toxicity by Formulated Drug and Combinations of Commercial ASA and AAA in Wistar Strain Albino Rats

Wistar strain albino rats were acclimatised at 22 ± 3 °C with a relative humidity of 30–70% and fed with a standard diet of “Amrut” procured from Krishna Valley AgrotechLLP, Pune, Maharashtra, India, and water ad libitum. They were maintained for a 12:12 h light-dark cycle and divided into six groups of *n* = 6 as per approval from the institutional animal ethics committee (IASST/IAEC/2022/09). 

#### 5.8.1. Neutralisation of Hyperglycemia and Prolonged Tail Bleeding Time

The venom yield for an adult *M. tamulus* is 1.5 mg [77]. Therefore, an average 60 kg adult can receive a maximum of 1.5 mg MTV in a sting equivalent to 25 µg/kg (or 5 µg/200 g of rat) 0.5 µg/20 g for an adult human of 60 kg weight. Individual groups of Wistar strain albino rats (200 ± 10 g, *n* = 6) were injected intravenously with 25 µg of MTV (approximately five times higher than the amount of MTV injected in one sting to human). The venom was dissolved in 0.20 mL of 1XPBS and injected (i.v.) into venom-injected groups of rats (*n* = 6); the group of rats injected with 1XPBS served as a control. 

Wistar strain albino rats (*n* = 6) were injected (intravenously or i.v) with formulated drug (formulation 2), combinations of ASA and AAA [ASA (1500 µg): AAA (50 µM] and ASA (187.5 µg): AAA (3 µM)) as well as the individual component of the formulation simultaneously with MTV (25 µg/200 g or 125 µg/kg of rats). Blood glucose content (mg/dL) of each group was measured by an accu-chek active glucometer at 0 to 240 min (at an interval of 30 min) by tail prick method [78]. All the groups were observed for 24 h for any physical or behavioural change, viz., body weight, food and water intake, defecation and urination, grip strength, and death. 

After 24 h of observation, the tail bleeding time of each Wistar strain albino group treated with MTV and formulated drug/MTV(venom-treated group)/PBS (control group) was determined by transverse amputation of the rat tail tip followed by immersing the tail in saline at 37 °C and monitoring the bleeding time [78].

#### 5.8.2. Neutralisation of Changes in Serum Biochemical Parameters

After 24 h post-injection, rats were sacrificed using chloroform (1%), and blood was collected immediately by cardiac puncture. The serum was isolated by centrifugation of the blood samples at 2200 rpm at 4 °C for 15 min. Biochemical parameters of blood serum viz. alkaline phosphatase (ALKP), serum glutamic pyruvic transaminase (SGPT), and creatinine were analysed by BeneSphera C61 semi-automatic biochemistry analyser using commercial diagnostic kits following the manufacturer’s instructions.

#### 5.8.3. Neutralisation of Morphological Alterations in Vital Organs 

The kidney, heart, liver, lung, testis, and ovary of treated and control groups were dissected 24 h post-observation to determine the possible morphological alterations. Tissues were cut and washed extensively in 1XPBS to remove the adhered blood clots and fixed in 10% buffered formaldehyde. The fixed tissues were dehydrated in grade concentrations of alcohol and embedded in paraffin wax. Section 4 and Section 5 mm thick were processed routinely for light microscopic observation after hematoxylin-eosin staining (H&E) for pathological studies [33,79].

### 5.9. Determination of MTV-Induced Inflammatory Cytokines Levels in Swiss Albino Mice

Swiss albino mice (22–24 g) were divided into two groups (control and treated). Group I (*n* = 6) mice were injected with 200 µL 1X PBS and considered a control. A group of mice injected (intravenously or i.v) with 25 µg of MTV dissolved in 1X PBS in the tail vein were considered as treated. Mice were maintained and used according to animal welfare international recommendations.

At 24 h of venom injection, blood was collected by cardiac puncture, and the level of blood plasma pro-inflammatory cytokines was determined. According to the manufacturer’s guide, the plasma levels of pro-inflammatory cytokines (IL-1β, IL-6, and TNFα) were assayed by ELISA using the mice-specific immunoassay kits (R&D Systems, Minneapolis, MN, USA). 

### 5.10. Statistical Analysis

The significance of the difference for more than two sets of data was analysed by a one-way ANOVA in Graphpad Prism 5, and the significant difference between two sets of data was determined by Student’s *t*-test analysis, using Sigma Plot 11.0 for Windows (version 10.0). When the *p*-value between the two sets of data was ≤0.05, then it was considered to be a significant difference.

## 6. Patent

Mukherjee, A.K., Das, B., and Khan, M.R. Indian Patent on “A pharmaceutical composition for the treatment of Indian red scorpion venom (*Mesobuthus tamulus*) induced toxicity” Patent application no. 202331027004.

## Figures and Tables

**Figure 1 toxins-15-00504-f001:**
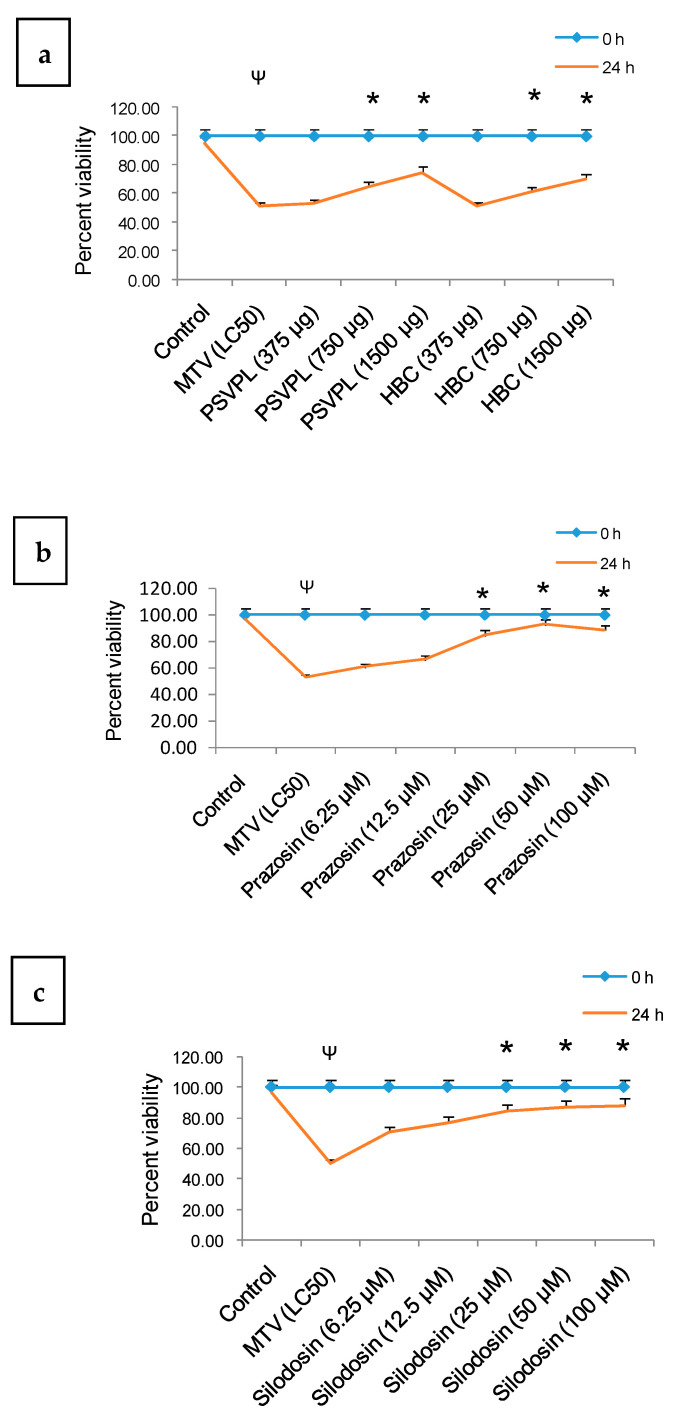
Dose-dependent in vivo neutralisation/inhibition of MTV (LC_50_)-induced toxicity in *C. elegans* at different time intervals by (**a**) commercial ASAs (375–1500 µg), (**b**) Prazosin (6.25 µM–100 µM), (**c**) Silodosin (6.25 µM–100 µM), (**d**) Terazosin (6.25 µM–50 µM), and (**e**) ascorbic acid (1 µg/mL–10 µg/mL). Data represent ± SD of three determinations. Significance of difference between control and MTV, ^Ψ^
*p* ≤ 0.05; between MTV and the ASAs, * *p* ≤ 0.05. (Abbreviations: LC: lethal concentration; ASAs: anti-scorpion-antivenoms; HCL: hydrochloride; MTV: *M. tamulus* venom).

**Figure 2 toxins-15-00504-f002:**
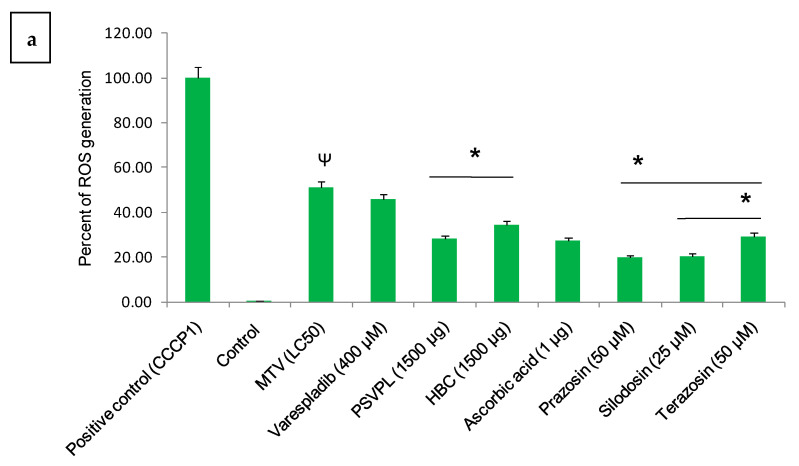
(**a**) Illustrates fluorescence intensities of MTV-induced ROS generation in *C. elegans* after 6 h of MTV (LC_50_) treatment and its neutralisation by commercial ASAs, Prazosin, ascorbic acid, Silodosin, and Terazosin determined by ImageJ 1.53j software. ROS level in the positive control (CCCP1) *C. elegans* was considered baseline (100%), and other values were compared with that. (**b**) Disruption of mitochondrial membrane potential (MMP) in MTV-treated *C. elegans* was observed after 24 h, and its neutralization by commercial ASAs, Prazosin, ascorbic acid, Silodosin, and Terazosin was determined with measurement of the ratio of red/green fluorescence intensity by JC-1 staining. Image J 1.53j software determined the image’s intensity, and the bar diagram plotted from the figures shown in Appendix A. Mitochondrial ROS level in the positive control (CCCP1) *C. elegans* was considered baseline (100%), and other values were compared. Data represent ± SD of three determinations. Significance of difference between control and MTV, ^Ψ^
*p* ≤ 0.05; between MTV and the ASAs/AAAs, * *p* ≤ 0.05. (Abbreviations: LC: lethal concentration; ROS: reactive oxygen species; ASAs: anti-scorpion-antivenoms; AAA: α1-adrenoreceptor antagonist; HCL: hydrochloride; CCCP1: Carbonyl cyanide 3-chlorophenylhydrazone 1).

**Figure 3 toxins-15-00504-f003:**
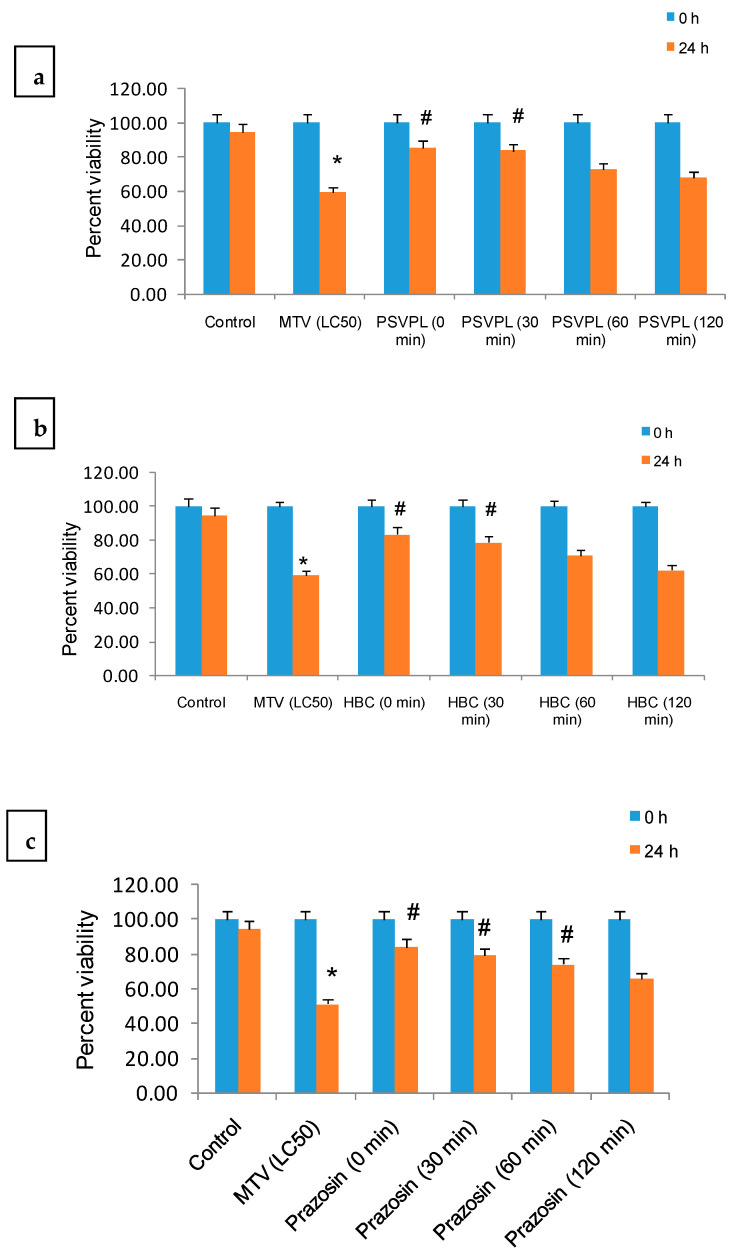
Neutralisation of MTV (LC_50_)-induced toxicity in *C. elegans* by an optimum dose of ASAs/AAAs (determined in Figure 1, Appendix A), which were added at an interval of 0–120 min post-treatment with venom. (**a**,**b**) PSVPL and HBC ASAs, (**c**) Prazosin, (**d**) Silodosin, and (**e**) Terazosin against MTV (LC_50_) in *C. elegans*. Data represent ± SD of three determinations. Significance of difference between control and MTV, * *p* ≤ 0.05; between MTV and the ASAs/AAAs, ^#^
*p* ≤ 0.05. (Abbreviations: LC: lethal concentration; ASAs: anti-scorpion-antivenoms; AAA: α1-adrenoreceptor antagonist; HCL: hydrochloride; PSVPL: Premium Serum and Vaccines Pvt. Ltd.; HBC: Haffkine Bio-pharmaceutical Corporation Ltd.; MTV: *M. tamulus* venom).

**Figure 4 toxins-15-00504-f004:**
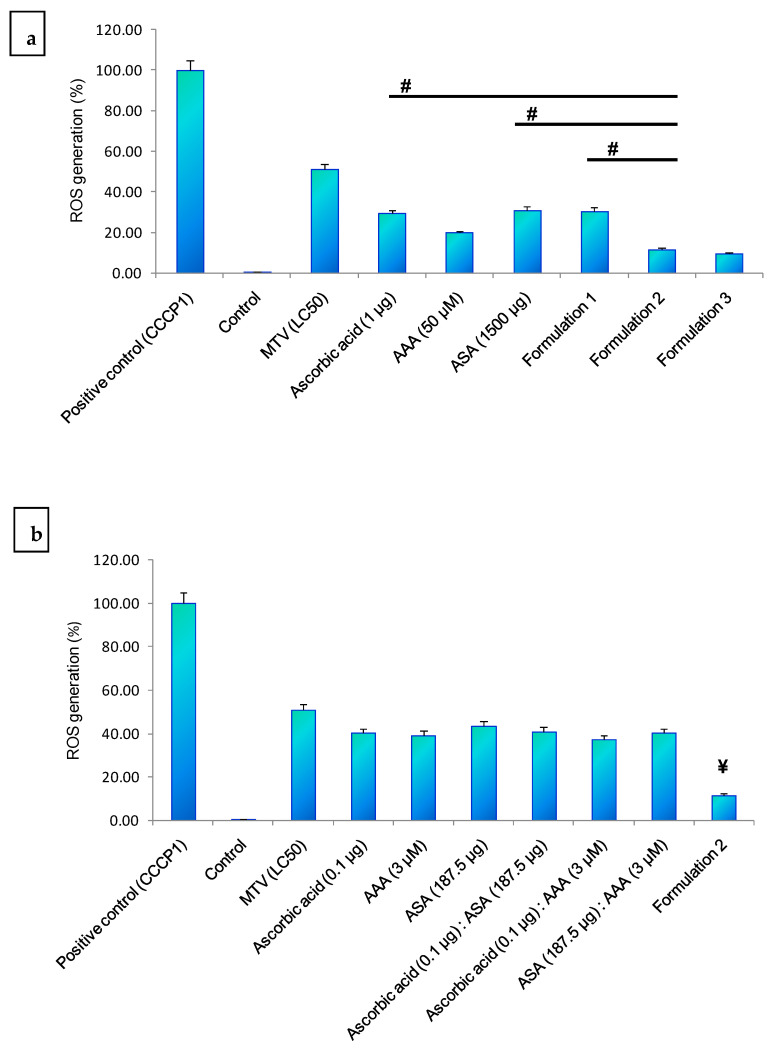
MTV-induced ROS generation in *C. elegans* after 6 h of MTV (LC_50_) treatment and its neutralisation. (**a**) Optimum dose of individual components of the formulation, their combinations, and different concentrations of formulations, (**b**) Individual components of the formulation and their combinations compared with formulation 2. (**a**) shows the significance of the difference compared to formulation 2, ^#^
*p* ≤ 0.05. (**b**) shows the significance of the difference compared to formulation 2, ^¥^
*p* ≤ 0.05. There was no significant difference (*p* > 0.05) between formulations 2 and 3. Fluorescence intensities were determined by ImageJ 1.53j software. Data represent ± SD of three Determination. (**c**) Fluorescence image of ROS generation (determined in confocal microscope) in *C. elegans* after 6 h of MTV (LC_50_ concentration) treatment and its neutralisation by formulation 2, individual components of the formulation, and their combinations. ROS level in the positive control (CCCP1) *C. elegans* was considered as baseline (100%) and other values were compared with that. (Abbreviations: LC: lethal concentration; ROS: reactive oxygen species; MTV: *M. tamulus* venom; CCCP1: carbonyl cyanide 3-chlorophenylhydrazone 1).

**Figure 5 toxins-15-00504-f005:**
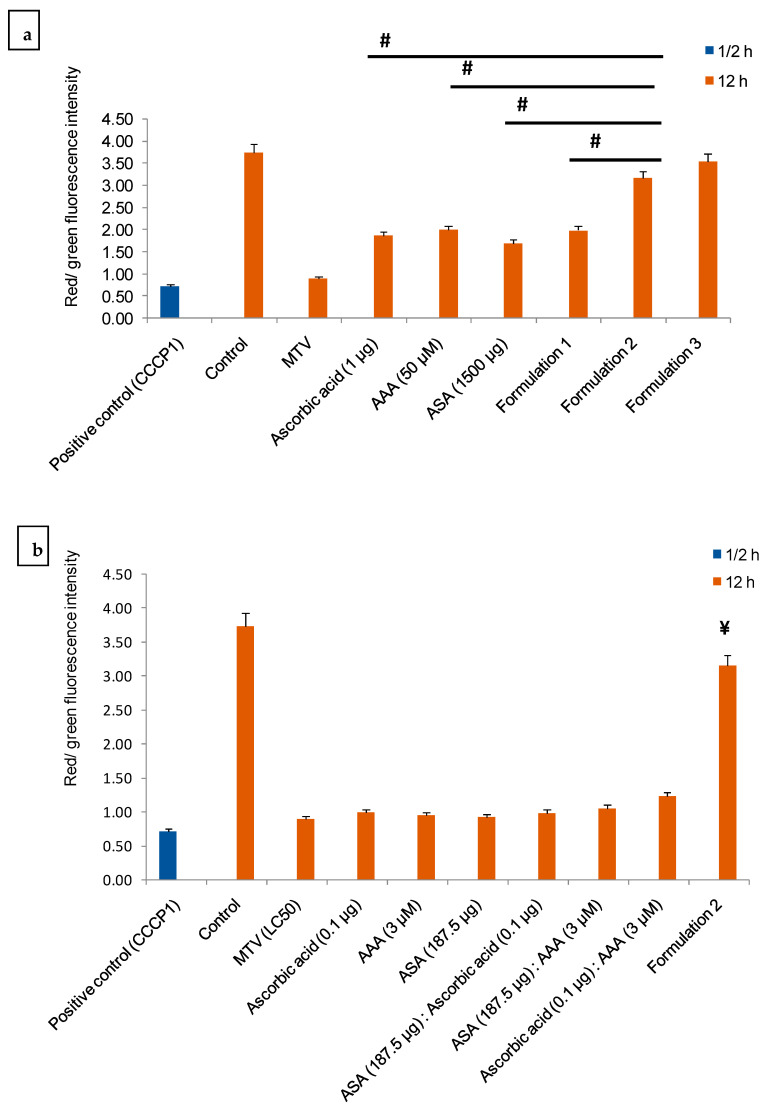
The in vivo neutralisation of MTV-induced (LC_50_ value) alteration of MMP in *C. elegans* by (**a**) Optimum dose of individual components of the formulation, their combinations, and different concentrations of formulation; (**b**) Individual components of the formulation and their combinations compared with formulation 2. Alteration of MMP induced by positive control (CCCP1) and MTV (LC_50_) in *C. elegans* was considered as baseline, and other values were compared. Fluorescence intensities were quantitated by ImageJ 1.53j software. Data represent ± SD of three Determination. Significance of difference, **^#^**
*p* ≤ 0.05 as compared to formulation 2. In (**a**) the significance of the difference compared to formulation 2, ^#^
*p* ≤ 0.05. (**b**) shows the significance of the difference compared to formulation 2, ^¥^
*p* ≤ 0.05. There was no significant difference (*p* > 0.05) between formulations 2 and 3. (**c**) Confocal microscopy images of MTV-induced alteration of MMP and its neutralization by formulation 2, individual components of the formulation, and their combinations. ROS level in the positive control (CCCP1) *C. elegans* was considered as baseline (100%), and other values were compared with that. (Abbreviations: LC: lethal concentration; CCCP1: carbonyl cyanide 3-chlorophenylhydrazone 1).

**Figure 6 toxins-15-00504-f006:**
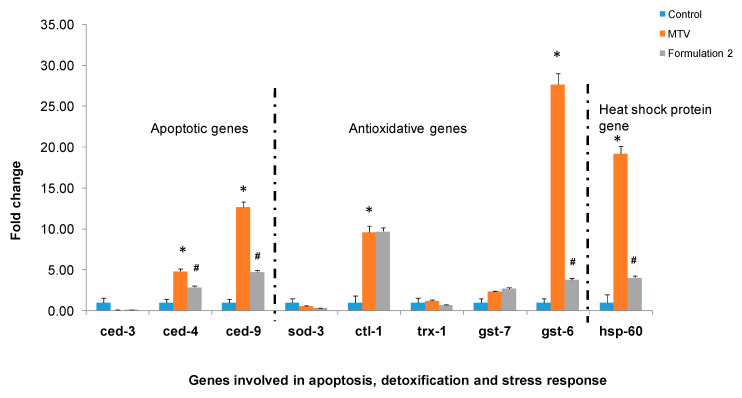
Demonstrated the relative expression of MTV-induced *C. elegans* genes involved in apoptosis, detoxification, and stress response compared to control (* *p* ≤ 0.05) and improvement by treatment with formulation 2. Significance of difference as compared to MTV (^#^
*p* ≤ 0.05) (Abbreviations: MTV: *M. tamulus* venom).

**Figure 7 toxins-15-00504-f007:**
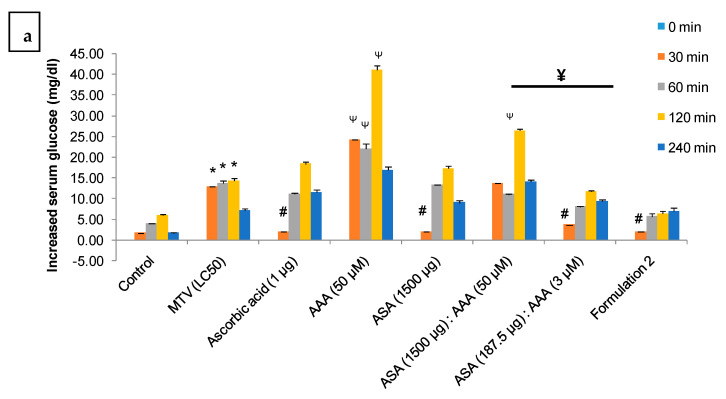
(**a**). Illustrates the time-dependent increase in blood glucose content in MTV-treated (25 µg/200 g, i.v.) Wistar strain albino rats and its neutralisation by formulation 2, individual component of the formulation at their optimum dose (the dose of individual component where they showed best MTV neutralisation (LC_50_ value) potency in *C. elegans*, (Figure 1b–d)) and combinations of commercial ASA and AAA against MTV (25 µg/200 g, i.v.) in Wistar rats. Significance of difference * *p* ≤ 0.05, as compared to control; ^#^
*p* ≤ 0.05, as compared to MTV; AAA (50 µM) and [AAA (50 µM): ASA (1500 µg] instead increase the blood glucose content in MTV treated rat, ^Ψ^
*p* ≤ 0.05. Significance of difference, ^¥^
*p* ≤ 0.05, between the combination doses [ASA (1500 µg): AAA (50 µM)] and [ASA (187.5 µg): AAA (3 µM)]. (**b**). Increase in tail bleeding time (sec) in MTV-treated (25 µg/200 g, i.v) Wistar strain albino rats and its neutralisation by formulation 2, individual components of the formulation at their optimum dose and combinations of commercial ASA and AAA against MTV (25 µg/200 g, i.v.) in Wistar rats. Data represent ± SD of three determinations. Significance of difference ^#^
*p* ≤ 0.05 compared to formulation 2. The significance of difference, ^¥^
*p* ≤ 0.05, between the combination doses [ASA (1500 µg): AAA (50 µM)] and [ASA (187.5 µg): AAA (3 µM)]. (Abbreviations: LC: lethal concentration; ASAs: anti-scorpion-antivenoms; AAA: α1-adrenoreceptor antagonist; MTV: *M. tamulus* venom).

**Table 1 toxins-15-00504-t001:** Co-treatment with ascorbic acid and commercial ASA at different time intervals against LC_50_ concentration of MTV in *C. elegans* (*n* = 50). Data represent ± SD of three determinations (*n* = 50). Significance of difference between control and MTV, ^Ψ^
*p* ≤ 0.05; between MTV and the ASAs, * *p* ≤ 0.05.

S. No.	Components	Viability of *C. elegans* (%)	
		0 h	24 h
1	Control (untreated)	100.00 ± 5	96.30 ± 4.8
2	MTV (LC_50_ value, 125 µg/mL)	100.00 ± 5	51.92 ± 2.6 (^Ψ^)
3	MTV co-treated with ascorbic acid (0 min): ASA (60 min)	100.00 ± 5	77.19 ± 3.8 (*)
4	MTV co-treated ascorbic acid (0 min): ASA (120 min)	100.00 ± 5	69.49 ± 3.47 (*)

**Table 2 toxins-15-00504-t002:** The in vivo neutralization of MTV (LC_50_ value)-induced toxicity in *C. elegans* with different concentrations of the formulated drugs, individual components of the formulations, and their combinations. Data represent mean ± SD of three determinations (*n* = 50). The significant difference between control and MTV, ^Ψ^
*p* ≤ 0.05; Significance of difference between formulation 1 and 2, * *p* ≤ 0.05.

S. No.	Components	Viability of *C. elegans* (%)	
		0 h	24 h
1	Control	100.00 ± 5	91.5 ± 4.51
2	MTV (LC_50_ value, 125 µg/mL)	100.00 ± 5	51.6 ± 2.57 (^Ψ^)
3	MTV (LC_50_ value) pre-treated with ASA (187.5 µg)	100.00 ± 5	59.5 ± 2.97
4	MTV (LC_50_ value) treated with AAA (Prazosin, 3 µM)	100.00 ± 5	64.4 ± 3.22
5	MTV (LC_50_ value) treated with ascorbic acid (0.1 µg)	100.00 ± 5	63.2 ± 3.16
6	MTV (LC_50_ value) treated with ASA (187.5 µg): AAA (3 µM)	100.00 ± 5	58.9 ± 2.94
7	MTV treated with ascorbic acid (0.1 µg): AAA (3 µM)	100.00 ± 5	63.2 ± 3.11
8	MTV treated with ASA (187.5 µg): ascorbic acid (0.1 µg)	100.00 ± 5	64.5 ± 3.22
9	MTV treated with ASA (93.75 µg): ascorbic acid (0.05 µg): AAA (1.5 µM) [Formulation 1]	100.00 ± 5	75.0 ± 3.75
10	MTV treated with ASA (187.5 µg): ascorbic acid (0.1 µg): AAA (3 µM) [Formulation 2]	100.00 ± 5	82.6 ± 4.13 (*)
11	MTV treated with ASA (375 µg): ascorbic acid (0.2 µg): AAA (6 µM) [Formulation 3]	100.00 ± 5	89.1 ± 4.45

**Table 3 toxins-15-00504-t003:** Neutralisation of MTV (25 µg/200 g, i.v)-induced elevation in ALKP, SGPT, and creatinine level in the blood of rats (24 h post-injection) by formulation 2, individual components of the formulation at their optimum dose (the dose of individual component where they showed best MTV neutralisation (LC_50_ value) potency in *C. elegans*, (Figure 1b–d)) and combinations of commercial ASA and AAA against MTV in Wistar rats (*n* = 6). Data represent mean ± SD of three determinations. The significant difference between control and MTV, ^Ψ^
*p* ≤ 0.05; Significance of difference between MTV and the other components of formulation 2, * *p* ≤ 0.05.

S. No.	Components	ALKP Activity (U/L)	SGPT Activity (U/L)	Creatinine Activity (U/L)
1	Control	108.00 ± 5.13	48.11 ± 2.40	0.27 ± 0.01
2	MTV (LC_50_)	180.50 ± 5.02 (^Ψ^)	75.77 ± 3.78 (^Ψ^)	0.44 ± 0.02 (^Ψ^)
3	ASA (1500 µg)	122.20 ± 3.45 (*)	39.90 ± 1.20 (*)	0.31 ± 0.01 (*)
4	Ascorbic acid (1 µg)	113.00 ± 5.31 (*)	45.66 ± 2.28 (*)	0.29 ± 0.01 (*)
5	AAA (50 µM)	121.50 ± 4.56 (*)	40.75 ± 2.03 (*)	0.33 ± 0.02 (*)
6	ASA (1500 µg): AAA (50 µM)	106.90 ± 4.96 (*)	42.00 ± 2.13(*)	0.30 ± 0.01 (*)
7	ASA (187.5 µg): AAA (3 µM)	130.00 ± 3.12 (*)	51.00 ± 2.55 (*)	0.39 ± 0.01 (*)
8	Formulation 2	102.71 ± 4.20 (*)	37.00 ± 1.85 (*)	0.28 ± 0.01 (*)

## Data Availability

Data is contained within the article and the Appendix A.

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
