# Peer review of "A Novel Therapeutic Formulation for the Improved Treatment of Indian Red Scorpion (Mesobuthus tamulus) Venom-Induced Toxicity-Tested in Caenorhabditis elegans and Rodent Models"

_toxins, 2023, doi:10.3390/toxins15080504_

Round 1

Reviewer 1 Report

Presented work is very well conceived and rightly persuaded research in a duly needed area related to scorpion envenomation and therapeutic development.  The study is very comprehensively conducted and presented and "par excellence" thus deserved to be published in Toxins. 

A minor suggestion: a huge data set is presented in supplementary file. As the use of C. elegans toxicity model in venom research is new and seems examplary, worth adding one figure in main MS text - e.g., effect of optimum formulation 2 (Fig. S6 - S7).     

Author Response

Reply to Reviewer 1

Comments to authors:

General comment: Presented work is very well conceived and rightly persuaded research in a duly needed area related to scorpion envenomation and therapeutic development. The study is very comprehensively conducted and presented and "par excellence" thus deserved to be published in Toxins.

Reply: Thank you for your valuable suggestions and encouraging words. We have tried to address your concerns in the revised manuscript.

Query: A minor suggestion: a huge data set is presented in supplementary file. As the use of C. elegans toxicity model in venom research is new and seems examplary, worth adding one figure in main MS text - e.g., effect of optimum formulation 2 (Fig. S6 - S7).

Reply: Thank you for your suggestion. Fig. S6-S7 has added in the main MS text (Page no. 16-24).

Reviewer 2 Report

I read the paper entitled "A therapeutic formulation for the improved treatment of Indian red scorpion (Mesobuthus tamulus) venom-induced toxicity tested in Caenorhabditis elegans and rodent models" submitted to the journal Toxin. The work describes a study carried out on C. elegans and on rats regarding the search for an adequate pharmaceutical formulation for the therapy of poisonings produced by scorpion bites, since the administration of conventional anti-venom antibodies alone does not seem to be completely effective due to the low presence of neutralizing antibodies in these preparations. Therefore, over the years neutralizing antibodies have been associated with other drugs such as alpha-adrenergic receptor antagonists (Prazosin) that have significantly improved the prognosis of poisoning, and the administration of antioxidants such as Vit C. In the study described, the efficacy of a formulation composed of an association of neutralizing antibodies, alpha-adrenergic receptor antagonist drugs, and ascorbic acid in poisoning C. elegans and rats is evaluated. The results seem to indicate a better efficacy of the pharmaceutical formulation in improving the prognosis compared to the effects of the individual components of the formulation itself. Moreover, This study demonstrates for the first time that C. elegans can be a model organism for screening the neutralization potency of the drug molecules against a neurotoxic scorpion venom.

I have many concerns regarding this paper

An animal model is used if its physiology is very close to that of humans to try to understand the physiology of the latter, or if the study of a physiological response can provide knowledge on how this has developed in the course of evolution. A fortiori, an animal model of a human disease must be as close as possible to humans to understand how the pathophysiology of the disease develops through the dysfunction of interrelated organs and structures. If this is not considered, there is the risk of using an animal model as a simple aggregate of cells, losing the supracellular aspect (of organ, tissue, and system) that characterizes the pathophysiology of a disease. The poison of the Indian red scorpion is composed of numerous toxins. Some are effective on Potassium SK channels, others on Sodium channels (Pedarzani et al, 2002). Others are antagonists of alpha-adrenergic receptors. Intoxication in humans seems to be characterized by autonomic and cardiovascular effects with a first phase of transient cholinergic hyperfunction, followed by a phase of dose-dependent adrenergic hyperactivity characterized by hypertension, tachycardia, and alteration of myocardial function. The latter seems to depend on the increase in the entry of calcium into cardiac myocytes, so Nifedipine has been used over time to "protect" the myocytes themselves. On the basis of clinical observations, therefore, therapeutic associations have been attempted to the necessary administration of neutralizing antibodies to the toxin. These associations, if they are to be studied at the level of laboratory animals, require an animal model with a physiology similar to that of humans, since the observed alterations are resolved mainly in cardiovascular alterations and in the consequences of the altered perfusion of the various organs resulting from them. C. elegans can be an interesting model if we observe for example the same effects of Tamapin on the membrane channels of excitable cells. I understand the binding of the toxin to the receptor which is similar to the alpha-adrenergic receptor of C-elegans, but the question I ask myself is: what is the physiological effect of stimulating this receptor? This is my (physiological) response that I expect altered by the poison. ROS increase as a result of poisoning. This does not surprise me considering the alterations in tissue perfusion resulting from cardiovascular effects. A similar situation, clinically, is observed in sepsis. Also in sepsis is used as a digger ascorbic acid. What kind of similarity is there between sepsis and ROS alterations observed in scorpion poisoning? This is the question that the doctor immediately asks, with the thought of being able to take advantage of perhaps using therapeutic approaches against free radicals that are formed in sepsis.

 Other major concerns regard the statistical analyses. Which kind of statistical tests are used? Maybe are written in the legends or in between the text. I would add a paragraph in the Material and Methods section. How many worms are utilized in each replicate? Or each replicate is an analysis of the same sample derived from the same worms? Numerosity is lacking in some Figures.

Therefore I suggest rejecting the paper encouraging Authors to improve experiments and the analytical approach of the study

No comments

Author Response

Reviewer 2 comments to authors:

General comment: I read the paper entitled "A therapeutic formulation for the improved treatment of Indian red scorpion (Mesobuthus tamulus) venom-induced toxicity tested in Caenorhabditis elegans and rodent models" submitted to the journal Toxin. The work describes a study carried out on C. elegans and on rats regarding the search for an adequate pharmaceutical formulation for the therapy of poisonings produced by scorpion bites, since the administration of conventional anti-venom antibodies alone does not seem to be completely effective due to the low presence of neutralizing antibodies in these preparations. Therefore, over the years neutralizing antibodies have been associated with other drugs such as alpha-adrenergic receptor antagonists (Prazosin) that have significantly improved the prognosis of poisoning, and the administration of antioxidants such as Vit C. In the study described, the efficacy of a formulation composed of an association of neutralizing antibodies, alpha-adrenergic receptor antagonist drugs, and ascorbic acid in poisoning C. elegans and rats is evaluated. The results seem to indicate a better efficacy of the pharmaceutical formulation in improving the prognosis compared to the effects of the individual components of the formulation itself. Moreover, this study demonstrates for the first time that C. elegans can be a model organism for screening the neutralization potency of the drug molecules against a neurotoxic scorpion venom.

Reply: Thank you for your precise time going through the article and your valuable suggestions to increase the quality of work. Thanks for understanding that this is the first report showing for the first time that C. elegans can be an effective model organism for screening the neutralization potency of the drug molecules against a neurotoxic scorpion venom. Throughout the globe, there is a growing apprehension about minimizing the use of experimental animals, although pre-clinical studies are indispensable. Therefore, our effort can save the life of several experimental animals, which is the most significant of this work. Please be informed that we have tried to address all the reviewer's concerns.

Query: I have many concerns regarding this paper.

An animal model is used if its physiology is very close to that of humans to try to understand the physiology of the latter, or if the study of a physiological response can provide knowledge on how this has developed in the course of evolution. A fortiori, an animal model of a human disease must be as close as possible to humans to understand how the pathophysiology of the disease develops through the dysfunction of interrelated organs and structures. If this is not considered, there is the risk of using an animal model as a simple aggregate of cells, losing the supracellular aspect (of organ, tissue, and system) that characterizes the pathophysiology of a disease. The poison of the Indian red scorpion is composed of numerous toxins. Some are effective on Potassium SK channels, others on Sodium channels (Pedarzani et al, 2002). Others are antagonists of alpha-adrenergic receptors. Intoxication in humans seems to be characterized by autonomic and cardiovascular effects with a first phase of transient cholinergic hyperfunction, followed by a phase of dose-dependent adrenergic hyperactivity characterized by hypertension, tachycardia, and alteration of myocardial function. The latter seems to depend on the increase in the entry of calcium into cardiac myocytes, so Nifedipine has been used over time to "protect" the myocytes themselves.

On the basis of clinical observations, therefore, therapeutic associations have been attempted to the necessary administration of neutralizing antibodies to the toxin. These associations, if they are to be studied at the level of laboratory animals, require an animal model with a physiology similar to that of humans, since the observed alterations are resolved mainly in cardiovascular alterations and in the consequences of the altered perfusion of the various organs resulting from them. C. elegans can be an interesting model if we observe for example the same effects of Tamapin on the membrane channels of excitable cells. I understand the binding of the toxin to the receptor which is similar to the alpha-adrenergic receptor of C-elegans, but the question I ask myself is: what is the physiological effect of stimulating this receptor? This is my(physiological) response that I expect altered by the poison. ROS increase as a result of poisoning. This does not surprise me considering the alterations in tissue perfusion resulting from cardiovascular effects. A similar situation, clinically, is observed in sepsis. Also in sepsis is used as a digger ascorbic acid. What kind of similarity is there between sepsis and ROS alterations observed in scorpion poisoning? This is the question that the doctor immediately asks, with the thought of being ableto take advantage of perhaps using therapeutic approaches against free radicals that are formed in sepsis.

Reply: The authors sincerely thank the honorable reviewer for giving his time to review the manuscript. We also agree with the reviewer's view that "an animal model of a human disease must be as close as possible to humans to understand how the pathophysiology of the disease develops through the dysfunction of interrelated organs and structures." In our study, Caenorhabditis elegans have been used as a model organism, highlighting the neurotoxic effects of scorpion venom and the screening of the neutralization potency of the therapeutic formulation against neurotoxic scorpion venom. We want to inform the honorable reviewer that different studies throughout the globe have well demonstrated that C. elegans is a good model organism for studying the neurotoxic effect of neurotoxic compounds (such as pesticides) and their in vivo neurotoxic mechanism [1; 2]. The manuscript's Introduction and Discussion sections highlight the reasons for selecting C. elegans as a model organism for this study. We hope that the honorable reviewer will agree to our view.  

In this study, we have shown that our formulated drug has successfully inhibited the pathophysiology associated with neurotoxic symptoms (free radicals scavenging, ROS inhibition, inhibition of depolarization of MMP) resulting from the action of Indian red scorpion venom. Further, nowhere in the manuscript have we claimed that C. elegans is a final model for establishing the therapeutic efficacy of drug molecules, instead of that our aim of the study is to find an alternative in vivo screening model to solve the ethical issue associated with the animal experiment and also to follow 3R model suggested by WHO (Replacement, Reduction, and Refinement of tested animals) to reduce the animal experiments. Further, the best-screened formulation was validated in rodent models where the formulated drug has shown superior efficacy compared to the individual components of the same drug to neutralize the toxicity of Indian red scorpion venom, thus establishing the suitability of C. elegans model to screen drugs against Indian red scorpion venom. 

The hyperactivity of alpha-adrenergic receptor induced by scorpion venom toxins resulted in cardiovascular effects with a first phase of transient cholinergic hyperfunction, followed by a degree of dose-dependent adrenergic hyperactivity characterized by hypertension, tachycardia, and alteration of myocardial function. Further, the alpha-adrenergic receptor activates the neurotransmitter norepinephrine and the neurohormone epinephrine, also known as catecholamines. 

For assessing the neurotoxicity exhibited by scorpion sting C. elgans can be a good model organism. However, we also agree with the reviewer that there are some limitations to using C. elegans as a model organism, such as studying scorpion sting-induced cardiovascular alterations. Therefore, this should be studied in depth in rodent models. This limitation is now discussed in the revised manuscript (page no. 30). In a nutshell, in our study, we have shown that for screening purposes to assess the inhibition of scorpion venom-induced neurotoxicity, C. elegans can be used as a model organism; however, further validation studies should be performed in a higher rodent model, and we did that. The revised manuscript now discusses this (Pages 29-30).

Our study showed that C. elegans contains SER6, a homolog of α1-adrenergic receptors of humans. SER-6 is an octopamine receptor, almost identical to mammalian α1-receptors, and its stimulation causes an octopaminergic signal that involves an array of neuropeptides that activate receptors and induces activation of c-AMP response in CNS and thus show pathophysiology associate with neurotoxicity such as ROS generation, and disruption of mitochondrial membrane potential [1; 2]. This fact is now discussed in the revised manuscript (Page no. 30).   

Sepsis is a multi-organ dysfunction condition characterized by hyper-inflammation, oxidative damage, hypercoagulation, tissue hypoperfusion and hypoxia, immunological suppression, and multiorgan malfunction.

Although the sepsis induced by scorpion sting cannot be assessed in C. elegans; as it has no developed multi-organ system; however, ROS generation and alteration of mitochondrial transmembrane potential, the primary causes of sepsis, can be determined in C. elegans. We have demonstrated that the formulated drug effectively inhibits ROS production and restore the disruption of MMP. For management of sepsis, along with the antivenom treatment, antibiotic treatment is suggested [3; 4; 5]. Further, few studies have reported that use of prazosin to mitigate the sepsis in scorpion stinged patients [3]. Furthermore, a histopathological study in rodent models has shown a deleterious effect of MTV in different organs of rats and its effective neutralization by formulated drug, much better than the individual components of the drug. The studies of marker enzyme in the serum of MTV-treated rat showed increase of the SGPT, ALKP, and creatinine indicating damage in liver and kidney. Our formulated drug restore the level of marker enzyme significantly higher than the individual componenet. But we agree that a detailed investigation into the mechanism of sepsis is utmost needed. The revised manuscript now discusses this fact (Page no. 30).

Query: Other major concerns regard the statistical analyses. Which kind of statistical tests are used? Maybe are written in the legends or in between the text. I would add a paragraph in the Material and Methods section. How many worms are utilized in each replicate? Or each replicate is an analysis of the same sample derived from the same worms? Numerosity is lacking in some Figures.Therefore I suggest rejecting the paper encouraging Authors to improve experiments and the analytical approach of the study

Reply: Thank you for your comments and for correcting us. Statistical tests such as t-tests and ANOVA have been used in our study. A paragraph on statistical analysis has been added in the material and method section of the manuscript (Page no. 41). 

In each replicate of the experiments, 50 worms were used, which is sufficient because most of the studies have used 25-30 worms in each experiment. For restoration of MTV-induced expression level of C. elegans genes using qRT-PCR (section 5.7), approximately 500 worms were used. Each replicate analyzes the different samples derived from different sets of worms. The number of worms used in each experiment has been explained in the material and method section of the revised manuscript.

References:

[1] Bargmann, C. I. Beyond the connectome: how neuromodulators shape neural circuits. Bioessays,  34(6):  458-465, 2012.

[2] Komuniecki, R., Harris, G., Hapiak, V., Wragg, R. and Bamber, B. Monoamines activate neuropeptide signaling cascades to modulate nociception in C. elegans: a useful model for the modulation of chronic pain? Invertebrate Neuroscience,  12):  53-61, 2012.

[3] Alavi, S. M. and Azarkish, A. Secondary bacterial infection among the patients with scorpion sting in Razi hospital, Ahvaz, Iran.), 2011.

[4] Ms, P., 2010. Cellulitis, necrotizing fasciitis, and subcutaneous tissue infections, in: Mandell, D., Bennett (Ed.), Principles and Practice of Infectious Diseases, pp. 1289-1312.

[5] Wheatley Iii, G. H., Wait, M. A. and Jessen, M. E. Infective endocarditis associated with a scorpion sting. The Annals of Thoracic Surgery,  80(4): 1489-1490, 2005.

Reviewer 3 Report

Remarks about the manuscript: 

The authors have presented an excellent article, "A therapeutic formulation for the improved treatment of Indian red scorpion (Mesobuthus tamulus) venom-induced toxicity tested in Caenorhabditis elegans and rodent models". The article focuses on the effect of the red scorpion toxins treatment strategy with various formulations of commercially available options. The article discusses the impact of toxins on post-treatment outcomes, using C. elegans and rodents as test subjects. The manuscript showcases a thoughtfully selected collection of experiments and corresponding discussions that are accompanied by clear and comprehensive explanations.

However, I have some suggestions here.

Please carefully review the explanation for each control.

The English language writing needs some corrections as many words are combined throughout the manuscript and extra spaces.

The present manuscript contained most of the data required for publication in the toxins journal; therefore, I will recommend it for publication in its current form. 

This manuscript (ID: toxins-2460079) can be accepted without revision.

The English language writing needs some corrections as many words are combined throughout the manuscript and extra spaces.

Author Response

Reply to Reviewer 3

Comments to authors:

The authors have presented an excellent article, "A therapeutic formulation for the improved treatment of Indian red scorpion (Mesobuthus tamulus) venom-induced toxicity tested in Caenorhabditis elegans and rodent models". The article focuses on the effect of the red scorpion toxins treatment strategy with various formulations of commercially available options. The article discusses the impact of toxins on post-treatment outcomes, using C. elegans and rodents as test subjects. The manuscript show cases a thoughtfully selected collection of experiments and corresponding discussions that are accompanied by clear and comprehensive explanations.

However, I have some suggestions here.

Please carefully review the explanation for each control.

The English language writing needs some corrections as many words are combined throughout the manuscript and extra spaces.

The present manuscript contained most of the data required for publication in the toxins journal; therefore, I will recommend it for publication in its current form.

This manuscript (ID: toxins-2460079) can be accepted without revision.

Reply: Thank you for your suggestion for improving the manuscript and encouraging words. In our study, the ROS level in the positive control (CCCP1 treated) C. elegans was considered baseline, and other values were compared to that. On the other hand, control C. elegans were treated with only an NGM buffer explained in the manuscript. Technical errors with the English words and extra spaces have been corrected throughout the manuscript.

Round 2

Reviewer 2 Report

I read the revised version of the MS entitled "A therapeutic formulation for the improved treatment of Indian red scorpion (Mesobuthus tamulus) venom-induced toxicity tested in Caenorhabditis elegans and rodent models" submitted to the journal Toxin and Authors reply to my concerns.

My objections concerned the way in which observations from different animal models used are often translated to man without giving the reader the opportunity to reason about the different and sometimes distant physiologies of the models used with respect to man himself. Different anatomy, physiologies and, in behavioral experiments, different ecophysiologies, do not mean denying the scientific interest of the collected data, but rather giving them the way to be reasoned and thought by discussing them in the light of how those different physiologies can have evolved and how they can be correctly inserted into the evolutionary context.
Therefore, do not uncritically accept the data obtained from an experimental model on the simple basis that this has been used by others previously, but frame them and explain them to the reader in the evolutionary context of the animal model used.
In the revised form of the manuscript, it is my opinion that these considerations have now been inserted, helping the reader to a critical reading of the experimental data obtained